# Assessment of Bioactive Compounds under Simulated Gastrointestinal Digestion of Bee Pollen and Bee Bread: Bioaccessibility and Antioxidant Activity

**DOI:** 10.3390/antiox10050651

**Published:** 2021-04-23

**Authors:** Volkan Aylanc, Andreia Tomás, Paulo Russo-Almeida, Soraia I. Falcão, Miguel Vilas-Boas

**Affiliations:** 1Centro de Investigação de Montanha (CIMO), Instituto Politécnico de Bragança, Campus de Santa Apolónia, 5300-253 Bragança, Portugal; volkan@ipb.pt (V.A.); tomas@ipb.pt (A.T.); sfalcao@ipb.pt (S.I.F.); 2Laboratório Apícola–LabApis, Departamento de Zootecnia, Universidade de Trás-os-Montes e Alto Douro (UTAD), 5300-801 Vila Real, Portugal; prusso@utad.pt

**Keywords:** bee pollen, bee bread, bioactive compounds, in vitro digestion, bioaccessibility, antioxidant activity

## Abstract

Bee pollen and bee bread have always been regarded as excellent natural resources for application in food and pharmaceutical fields due to their rich nutrient content and diversity of bioactive compounds with health-improving properties. Extensive studies on both bee products as ingredients for a healthy diet were reported, although the data concerning their metabolization on the gastrointestinal tract is quite limited. Here, we report, at each digestive stage, the bioactive profile for both bee products, their bioaccessibility levels and the antioxidant activity evaluation. The findings indicated that the average bioaccessibility level of total phenolic and total flavonoid content for bee pollen was 31% and 25%, respectively, while it was 38% and 35% for bee bread. This was reflected in a decrease of their antioxidant capacity at the end of in vitro gastrointestinal digestion, both in free radicals scavenging capacity and in reducing power. Moreover, within the 35 phytochemicals identified, the most affected by gastrointestinal digestion were phenylamides, with a complete digestibility at the end of the intestinal phase. Overall, our results highlight that bioactive compounds in both raw products do not reflect the real amount absorbed in the intestine, being bee bread more accessible in bioactive content than bee pollen.

## 1. Introduction

In recent years, there has been an increasing interest in natural products for a balanced and healthy diet accrue from concerns with the synthetic food or food additives. In this context, there has been a growing interest from research to find natural functional products that have high nutritional value and positive effects on health, which includes bee products such as bee pollen and bee bread [1].

Bee pollen is formed from the combination of the pollen collected from flowers by bees, which they add nectar and their own secretions [2]. After returning to the hive, bee pollen is stored in the comb cells, mixed with organic acids, honey and digestive enzymes secreted by bees, undergoing a lactic acid fermentation process and transforming it into a new product called “bee bread” [1]. Bee bread differs from fresh bee pollen, mostly due to the changes, which occur during the fermentation and are related to the stability and digestibility of this bee product. Bee pollen and bee bread are characterized by their nutritive richness, which is based on the essential nutrients and phytochemicals of pollen, mainly carbohydrates, proteins and phenolic compounds [3,4]. The chemical composition of these bee products can vary according to the plant origin, the nutrient status of the plant, the geographical region conditions, the soil and factors such as the collection, storage and degree of processing [2]. Besides, these variabilities often affect the diversity and amount of the phytochemicals, such as the phenolic compounds responsible for their biological activities and consequently resulting in different therapeutic properties [1]. The studies in bee pollen and bee bread available in the literature have mostly focused on physicochemical properties [3,5], botanical origin [4,6], microbiological evaluation [7,8], phenolic compound content [9,10] and antioxidant activities [5,11].

The protective role of phytochemicals that are part of our diet has become an increasingly important area of human nutrition research [1,4,12]. Although most of the biological effects of bioactive compounds are attributed to their high antioxidant potential [5], it is now known that these compounds are more than just antioxidants: there is increasing evidence that long-term modest intakes of bioactive compounds may show the potential to limit the risk of degenerative diseases by protecting cells from oxidative damage [12]. Nevertheless, the simple uptake of rich phytochemical diets does not ensure its effectiveness since these compounds may be affected by the human gastrointestinal tract in different ways, and consequently, this may be reflected in their biological activity.

Hence, it is crucial to determine the changes due to bioactive compounds being exposed to the gastrointestinal tract. Although studies on the digestibility of both bee products are quite limited, there are a few papers, especially on bee pollen. Yesiltas et al. [13] determined the phenolic content and antioxidant activities of bee pollen and propolis using a gastrointestinal digestion model. Benavides-Guevara et al. [14] investigated the effect of different pretreatment applications, including enzymatic treatment, on the digestibility of bee pollen. Moreover, bee bread is reported to be more digestible than bee pollen [1,5,8], but to date, there has been no digestibility study of bee bread and also no comprehensive comparative study between bee pollen and bee bread in terms of the bioactive compounds following a simulated gastrointestinal digestion model. Therefore, this study reports the comparison of the bioavailability properties of bee pollen and bee bread using an in vitro simulated digestive model: digestibility and bioaccessibility level of bioactive compounds and their antioxidant capacities.

## 2. Materials and Methods

### 2.1. Standards and Reagents

Ethanol, sodium phosphate, potassium phosphate, acetonitrile, sodium hydroxide, trichloroacetic acid, hydrochloric acid, gallic acid, potassium ferricyanide, ferric chloride and calcium chloride dihydrate were purchased from Fisher Scientific (Pittsburgh, PA, USA). Folin–Ciocalteu’s reagent, potassium chloride, ammonium carbonate, acetic acid glacial and sodium chloride were purchased from Panreac Applichem (Barcelona, Spain). Magnesium chloride hexahydrate and aluminum chloride were from Acros Organics (Pittsburgh, PA, USA), and sodium carbonate anhydrous was purchased from Labkem (Barcelona, Spain).

Human salivary α-amylase (A1031-1KU), porcine pepsin (P6887), porcine pancreatin 4 × USP specifications (P1750), bile bovine (B3883), 2,2-diphenyl-1-picrylhydrazyl (DPPH), quercetin, chrysin and *p*-coumaric acid were purchased from Sigma-Aldrich (St. Louis, MO, USA). Kaempferol was purchased from Extrasynthese (Genay, France) and naringenin from Acros Organics (Pittsburgh, PA, USA).

### 2.2. Sample Collection and Preparation

Three different bee pollen and bee bread samples were collected in August 2019 from *Apis mellifera iberiensis* hives located in different apiaries, in the northeast of Portugal. The bee pollen samples, coded as BP-A1, BP-A2 and BP-A3, were collected from the northwest of Bragança (Poulão), southeast of Bragança (Pinheiro Manso) and Vila Real, respectively, while the bee bread samples, coded as BB-A1, BB-A2 and BB-A3, were collected from the northwest of Bragança (Poulão), southeast of Bragança (Pinheiro Manso) and Miranda do Douro (Póvoa), respectively. The pollen samples were collected using pollen traps, while combs with bee bread were removed from inside of the beehives. Within the laboratory facilities, bee pollen samples were cleaned from debris of wood and dead bee parts, while combs were crushed manually to extract bee bread. All samples were freeze-dried and stored at −20 °C for further analysis.

### 2.3. Palynological Analysis

The homogenized sample, about 1 g, were placed in separate 50 mL Falcon tube with distilled water, followed by vigorous stirring to disrupt pollen and allow a representative subsample. Then, 200 μL were taken from the resulting mixture and centrifuged at 1000× *g* for 5 min. The obtained pellet was subjected to acetolysis according to the method reported previously [15]. Pollen identification and counting were performed using an optical microscope. More than 1200 grains per preparation were counted following the criteria of Vergeron [16].

### 2.4. Phenolic Compounds Extraction

The extraction was carried out according to the method reported by Tomás et al. [4]. Briefly, bee pollen and bee bread samples were powdered using a lab type blender and 2 g of each sample was mixed with 40 mL of EtOH/H_2_O (80:20, *v*/*v*), then stirred at room temperature for 6 h. The resulting mixture was filtered through a Whatman No. 4 filter paper and the residue was re-extracted under the same conditions. After, the extracts were combined and the solvent was evaporated at 40 °C in a rotavapor (Rotary Evaporator model Hei-VAP from Heidolph, Schwabach, Germany). Finally, bee pollen and bee bread extracts were freeze-dried using a lyophilizer (FreeZone 4.5 model 7750031 from Labconco, Kansas City, KS, USA) and stored at −20 °C until further analysis.

### 2.5. Phenolic Content

#### 2.5.1. Total Phenolic Content

Total phenolic content (TPC) was determined by the Folin–Ciocalteu method [17]. In the procedure, 0.5 mL of ethanolic extract (1 mg/mL) was mixed with 0.25 mL of Folin–Ciocalteu reagent. After 3 min, 1 mL of 20% Na_2_CO_3_ was added and the final volume adjusted to 5 mL with deionized water. The solutions were left in a water bath at 70 °C for 10 min and then cooled in the dark for 30 min. The absorbance was read at 760 nm using a spectrophotometer (Analytikijena 200–2004 spectrophotometer from Analytik Jena, Jena, Germany). The TPC value of the bee pollen and bee bread samples were expressed as milligram of gallic acid equivalent per gram of dry weight sample (mg GAE/g).

#### 2.5.2. Total Flavonoid Content

The total flavonoid content (TFC) was recorded spectrophotometrically according to Falcão et al. [17]. Briefly, 0.2 mL of ethanolic extract (5 mg/mL) was mixed to 0.2 mL of AlCl_3_ solution (2% AlCl_3_, in 5% glacial acetic acid, in methanol). Then 2.8 mL of 5% acetic acid/methanol was added to the mixture. After 30 min at room temperature, the absorbance was read at 415 nm using a spectrophotometer (Analytikijena 200–2004 spectrophotometer from Analytik Jena, Jena, Germany). The TFC value of the bee pollen and bee bread samples was expressed as milligram of quercetin equivalent per gram of dry weight sample (mg QE/g).

### 2.6. LC/DAD/ESI-MS^n^ Bioactive Compounds Analysis

For the analysis, bee pollen and bee bread phenolic extracts, and the soluble digestive fractions (20 mg) were dissolved in EtOH/H_2_O (80:20, *v*/*v*, 2 mL). All the samples were filtered through a 0.22 μm membrane filter and kept in the freezer at −20 °C, until analysis.

For the LC analyses, a Dionex UltiMate 3000 ultrapressure liquid chromatography instrument, coupled to a diode array detector (Thermo Fisher Scientific, San Jose, CA, USA) was used. The chromatographic column was a C18 column (250 mm × 4 mm id; 5 mm particles diameter; end-capped), from Macherey–Nagel Nucleosil: the temperature of the column was kept at 30 °C. The flow rate and the injection volume was 1 mL/min and 10 μL, respectively [9].

For the MS analysis, a LTQ XL linear ion trap mass spectrometer (Thermo Fisher Scientific, San Jose, CA, USA) with an ESI source operating in the negative ion mode was used. The ESI conditions were in accordance with the previously reported [9].

The phenolic compounds identification was achieved comparing the chromatographic performance, UV spectra and MS data with those of reference compounds. The structural information was confirmed by combining the UV results with the MS fragmentation data previously described in the literature, whenever the reference standards were not available. Calibrations curves for the following compounds were used in the quantification *p*-coumaric acid (0.00925–0.4 mg/mL; y = 1.9 × 10^7^x − 12,927; R^2^ = 0.996), quercetin (0.037–1.6 mg/mL; y = 4.0 × 10^6^x − 10,216; R^2^ = 0.997), kaempferol (0.037–1.6 mg/mL; y = 4.3 × 10^6^x − 13,567; R^2^ = 0.998), chrysin (0.0185–0.8 mg/mL; y = 1.2 × 10^7^x − 51,265; R^2^ = 0.999) and naringenin (0.0185–0.8 mg/mL; y = 8.0 × 10^6^x − 10,998; R^2^ = 0.998). If the standard was not available, the quantification was performed using the calibration curve of the standard structurally close, with the final result being expressed in equivalent terms, as mg/g of sample.

### 2.7. Antioxidant Activity

#### 2.7.1. DPPH Radical Scavenging Assay

DPPH free radical scavenging activity of samples was performed according to Tomás et al. [4] with some modifications. Of the phenolic extracts 0.15 mL, with concentrations ranging from 0.03 to 0.43 mg/mL were mixed with 0.15 mL of DPPH (50 mg/L) and the absorbance was read at 515 nm using an ELX800 Microplate Reader (Bio-Tek Instruments, Inc., Winooski, VT, USA). The percentage of radical inhibition was calculated using the following equation:$$\%\;\mathrm{Inhibition}=[(A_{\mathrm{DPPH}}-{\;A}_{\mathrm{Sample}})\;/{\;A}_{\mathrm{DPPH}}]\times 100$$

The amount of antioxidant necessary to decrease the initial DPPH concentration by 50% (EC_50_) was achieved plotting the inhibition percentage against the extract concentration.

#### 2.7.2. Reducing Power Assay

The assay was performed according to Falcão et al. [17]. Of ethanolic extract (1 mg/mL) 0.25 mL of the sample was mixed with 1.25 mL of phosphate buffer (0.2 M, pH 6.6) and 1.25 mL of 1% potassium ferricyanide, respectively. The mixture was left in a water bath at 50 °C for 20 min. Then, 1.25 mL of 10% trichloroacetic acid was added to the mixture and centrifuged at 3000× *g* (Centurion K2R series, Chichester, UK) for 10 min. Of the upper layer 1.25 mL was mixed with 1.25 mL of deionized water and 0.25 mL of 0.1% FeCl_3_, and the absorbance was read at 700 nm. Gallic acid was used as standard and the results were expressed as milligram of gallic acid equivalent per gram of dry weight sample (mg GAE/g).

### 2.8. In Vitro Gastrointestinal Digestion

The static in vitro digestion model was performed according to the method developed by the COST INFOGEST international network [18]. This method consists of three sequential phases: oral, gastric and intestinal digestion, Figure 1 and Appendix A. Briefly, 5 g of bee pollen or bee bread samples were mixed with 3.5 mL simulated saliva fluid stock solution. This step is followed by the addition of 0.5 mL α-amylase solution of 1500 U/mL. Then, pH was adjusted to 7 with 1 mol/L NaOH and incubated in a water bath, on the dark, at 37 °C for 2 min with constant shaking. In the gastric phase, the oral bolus was mixed with 7.5 mL of simulated gastric fluid stock solution followed by 1.6 mL pepsin solution of 25,000 U/mL. Then, pH was adjusted to 3 with 1 mol/L HCl, followed by an incubation during 2 h under the same conditions as in the oral phase. In the final phase, gastric chyme was mixed with 11 mL of simulated intestinal fluid stock solution, 5 mL of pancreatin solution of 800 U/mL, 2.5 mL of bile (160 mM in fresh bile) and NaOH (to adjust the pH 7). NaOH–HCl (both 1 mol/L) was used to set the pH back to 7 and incubated under the same conditions as the gastric phase. Finally, the obtained samples from the three digestion phases were centrifuged for 15 min at 10,000× *g* at 4 °C, the soluble and pellet fractions were stored at −32 °C until further analysis. Each bee pollen and bee bread sample was digested in triplicate and the replicates mixed.

### 2.9. Bioaccessibility

Bioaccessibility (%) was described as the amount of phenolic compounds released in the in vitro digestion process compared to the amount of phenolic compounds in the tested sample, and calculated according to the following equation [19]:$$\mathrm{Bioaccessibility}\left(\%\right)=\left(\frac{\mathrm{content}\;\mathrm{of}\;\mathrm{phenolic}\;\mathrm{compounds}\;\mathrm{released}\;\mathrm{in}\;\mathrm{the}\;\mathrm{simulated}\;\mathrm{digestion}}{\mathrm{content}\;\mathrm{of}\;\mathrm{phenolic}\;\mathrm{compounds}\;\mathrm{in}\;\mathrm{the}\;\mathrm{tested}\;\mathrm{sample}}\right)\times 100$$

### 2.10. Data Analysis

All analyses were performed in triplicate and the results were denoted as mean ± standard deviation (SD). The obtained data was analyzed using GraphPad Prism version 8 (San Diego, CA, USA). A one-way analysis of variance was performed, followed by Tukey’s test for mean separation at *p* < 0.05. Additionally, Pearson’s correlation coefficients were calculated to ascertain the relationship between the tested parameters.

## 3. Results

### 3.1. Botanical Origin of Bee Pollen and Bee Bread

Forty pollen types were identified, at the species, genus or family level, in the bee pollen and bee bread samples. To simplify data analysis, only those with relative frequency percentages, higher than 0.1%, are presented in Table 1. A total of 17 pollen types were found at percentages greater than 3%, resulting in a classification of all samples as heterofloral, since no major taxa were present at a relative frequency greater than 80% [2].

In the BP-A1 sample, *Plantago* sp. (47%) from Plantaginaceae family was the dominant pollen type, while *Crepis capillaris* (60%) from the Asteraceae family was the dominant pollen type in BP-A2. For the BP-A3 sample, *Cytisus striatus* (48%) from the Fabaceae family was the dominant taxa.

The presence of dominant taxon was not so evident for BB-A1 and BB-A3 samples. Instead, several different families had distribution at the accessory or isolated pollen level, with higher prevalence on *Cytisus striatus* (Fabaceae), *Castanea sativa* (Fagaceae), *Jasione montana* (Campanulaceae) and *Rubus* sp. (Rosaceae). *Castanea sativa* (48%) was the only dominant pollen type in the BB-A2 sample, followed by *Rubus* sp. with a relative frequency value of 22%. Asteraceae and Fabaceae families were described in previous studies as dominant pollens in bee pollen samples (*n* = 22) obtained from Douro International Natural Park in Portugal [3], which is in accordance to the results presently found. Besides, it is known that *Castanea* and *Rubus* pollen types are dominant in bee pollen and bee bread coming from the northern part of Portugal [4]. Even though the palynological results in the current study are in accordance with the works mentioned above, the pollen types in both bee products may vary depending on the collection season and apiary location [1].

### 3.2. Total Phenolic and Flavonoid Content, and Bioaccessibility Level

The polarities, molecular weights, differences in chemical structures and abundances of approximately eight thousand phytochemicals that exist in plants can affect in different ways the digestive process, and therefore their bioavailability may differ in the body [19]. Along with this, the total quantity of phenolic compounds in food matrices does not reflect the amount absorbed by humans. The in vitro digestion method is a simple, fast, cost-effective with no ethical concerns, simulation process that may provide important data on the stability of the bioactive compounds throughout the digestion process. Additionally, the obtained in vitro digestion model results show a good correlation with the data obtained from in vivo studies [20].

In the present study, the TPC and TFC of undigested (raw) and digested bee pollen and bee bread samples are illustrated in Figure 2. The TPC for raw bee pollen samples ranged from 2.4 ± 0.1 (BP-A2) to 4.3 ± 0.1 mg GAE/g (BP-A3), while raw bee bread samples ranged from 3.2 ± 0.2 (BB-A3) to 3.8 ± 0.1 mg GAE/g (BB-A2). The TFC in bee pollen and bee bread ranged from 0.6 ± 0.1 to 2.7 ± 0.1 mg QE/g, in the following order: BP-A3 > BB-A3 > BP-A1 > BB-A2 > BB-A1 > BP-A2. The variation observed in the results was correlated to the different botanical origin of the samples or the fermentation product in the case of the bee bread samples [1]. Additionally, the TPC for the bee bread samples were more homogeneous compared to bee pollen, although the TFC had highly variable values, with statistical difference between each sample (Figure 2A,C). These findings are consistent with previous studies indicating that bee pollen and bee bread are an important source of phenolic compounds with antioxidant activity [3,5,12].

After undergoing the oral phase, a slight decrease was observed in the TPC of the digested pollen samples, with values of 1.9 ± 0.1, 1.1 ± 0.1 and 3.9 ± 0.2 for BP-A1, BP-A2 and BP-A3 samples, respectively, Figure 2A,B. The gastric phase was clearly more shocking, with a decrease in TPC by 52% (BP-A1), 36% (BP-A2) and 69% (BP-A3). In the last phase of digestion, the intestinal, the TPC in BP-A1 maintained the decreasing profile and reached a minimum of 0.8 ± 0.1 mg GAE/g, while BP-A2 and BP-A3, revealed a slight increase, reaching 0.7 ± 0.1 and 1.5 ± 0.1 mg GAE/g, respectively. The TPC of bee bread samples showed a similar trend to bee pollen after oral and gastric phases, Figure 2B. However, a bigger increase was observed in TPC for all bee bread samples after the intestinal phase, and BB-A1, BB-A2 and BB-A3 reached 1.4 ± 0.0, 1.4 ± 0.1 and 1.1 ± 0.1 mg GAE/g, respectively.

According to the values reported in Figure 2C, the TFC showed a decreasing trend in the pollen samples after each digestive phase, except for BP-A1 and BP-A3 in the intestinal phase (Figure 2D). Additionally, for each pollen sample, there was no significant difference in TFC content between gastric and intestinal phases. On average, the TFC decrease in bee pollen samples was 77%, reaching 0.5 ± 0.1 (BP-A1), 0.2 ± 0.0 (BP-A2) and 0.5 ± 0.1 mg QE/g (BP-A3) at the end of digestion. A continuous decrease was also observed in the TFC for the bee bread samples. After the intestinal phase, the TFC in BB-A1, BB-A2 and BB-A3 reached 0.5 ± 0.0 mg QE/g, 0.6 ± 0.0 mg QE/g and 0.7 ± 0.1 mg QE/g, respectively.

In general, an overall decrease in the TPC of bee pollen and bee bread samples was observed at the end of the in vitro gastrointestinal digestion. Previous studies indicated that food matrices generally have a gradual decrease in their TPC as they pass through the digestive system [19,21], however, there are also studies reporting a slight increase in TPC at the end of digestion, despite a decrease in the oral and gastric phase for different foods [22,23], which was also observed in the current study, where the TPC of the samples was slightly increased in the intestinal phase compared to the gastric phase, Figure 2B. This may be related to the multilayered wall structure of the pollen grains, which is resistant to digestive enzymes and pH changes. Additionally, the porous structure of the pollen grains may have contributed to the continuous release of phenolic compounds. Another important point is the high acidity of the stomach environment, which will have a strong effect on the released phenolic compounds [21]. Besides, the higher phenolic content released from bee bread samples comparing to the bee pollen can be explained by the partial digestion of the multilayered structure of pollen grains by bacterial enzymes throughout the fermentation process of bee bread [8]. The findings in this study are in agreement with studies reported for TFC of different food matrices [13,21].

The TFC was generally found to be in a decreasing tendency, except for BP-A1 and BB-A3 samples in the intestinal phase (Figure 2D). This decrease may be attributed to the breakdown of released and more accessible flavonoids by the action of digestive enzymes or different pH environments [21]. In the study conducted by Pinto et al. [24], it was reported that there was a decrease in the TFC of elderberries at the end of the digestion. Similar results have been obtained by other studies on edible mushrooms and carob flour as well [25,26].

Based on the experimental results it was possible to calculate the bioaccessibility index for bee pollen and bee bread samples at the end of the digestion, as presented in Figure 3. The average TPC and TFC bioaccessibility levels for bee pollen samples were 31% and 25%, respectively, while it was 38% and 35% for bee bread samples. These differences in TPC and TFC of the samples resulted in different bioavailability scores of the total quantity of compounds to be absorbed at the end of digestion. According to the findings, it can be said that bee bread is a more digestible product and so, more accessible than bee pollen in terms of bioactive compounds.

### 3.3. Phytochemical Profile and Bioaccessibility Level

The individual phenolic compounds of both bee pollen and bee bread samples were investigated by LC/DAD/ESI-MS^n^, at the different digestion stages (Figure 4).

The chromatogram allowed the identification and quantification of 35 bioactive phytochemicals, which included 21 phenolic compounds (Table 2), mostly flavonols, and 14 phenylamides (conjugates of polyamines and hydroxycinnamic acids). The quantitative contribution of phenylamides was significantly higher than the other phenolic compounds.

Within the phenolic compounds, flavonol derivatives such as quercetin, kaempferol, isorhamnetin and herbacetin glycosides, where the main compounds in both bee pollen and bee bread samples. These flavonoids have also been observed in many reported studies for the phenolic content of both bee products and were used for confirming our results [6,9,10,28]. A few flavonoids such as kaempferol-*O*-rutinoside (*m*/*z* 593), quercetin-3-*O*-glucoside (*m*/*z* 463), quercetin-3-*O*-rhamnoside (*m*/*z* 447), hesperetin (*m*/*z* 301) and luteolin (*m*/*z* 285) were identified considering the retention time, UV–Vis profile and MS pattern of the commercial standards.

The identification of the other phytochemicals was done through interpretation of the fragmentation pathways detected in MS^n^ spectra, comparing with that available in the literature and combining with the spectral information from UV.

All the extracts typically contained flavonoid glycosides in their composition. The sugar moieties in the flavonoids were assigned to rutinosides, hexosides, glucosides and rhamnosides. These are the most common and frequent in nature and were also confirmed in these samples. Methyl herbacetin-*O*-dihexoside (*m*/*z* 639), isorhamnetin-*O*-pentosyl-hexoside (*m*/*z* 609), kaempferol-3-*O*-rutinoside (*m*/*z* 593) and quercetin-3-*O*-glucoside (*m*/*z* 463) were identified as the most common compounds in both bee pollen and bee bread samples. BP-A3 sample was the richest among all in terms of diversity of flavonoids, at the same time it had the highest flavonoid content with a value of 2.70 mg/g. Quercetin-3-*O*-rhamnoside (*m*/*z* 447), isorhamnetin-*O*-deoxyhexoside (*m*/*z* 461) and luteolin (*m*/*z* 447) were only assigned in the BP-A3 sample and were the major compounds contributing to its high flavonoid content. In bee bread samples, the flavonoid content was higher than BP-A1 and BP-A2, with a great diversity of substances in different amounts, particularly, herbacetin derivatives such as methyl herbacetin-*O*-rutinoside (*m*/*z* 623), methyl herbacetin-3-*O*-hexoside (*m*/*z* 447) and methyl herbacetin (*m*/*z* 315). This assignment was supported by previous identification of similar compounds in bee pollen and bee bread samples from *Brassicaceae* spp., *Asteraceae* spp., *Lavandula* spp. and *Plantago* spp. plant family [10,34,35].

This study enabled also the identification of an important group of compounds present in high concentrations in the bee pollen and bee bread samples, namely phenylamides and their derivatives. Phenylamides are low-molecular products of covalent bonding between carboxylic groups of hydroxycinnamic acids and amine groups of aliphatic di- and polyamines or aromatic monoamines [36]. In both, bee pollen and bee bread samples, the most widely distributed acidic parent compounds of phenylamides were caffeic, ferulic and *p*-coumaric acids, while the aliphatic polyamines spermidine and spermine were found as the predominant amine components of phenylamides. For the spermidines, the formation of the amide linkage between a phenylamide and the phenolic acid can occur in the *N^1^*, *N^5^* and *N^10^* positions [36]. Spermine are mostly found conjugated with coumaroyl moieties in the positions *N^1^*, *N^5^*, *N^10^* and *N^4^*. These polyamine conjugated with phenolic compounds have a predominance in the plant species of several families such as Fabaceae, Asteraceae, Amaryllidaceae and Araceae [37].

The content of phenylamides varied broadly between 7.9 and 38.7 mg/g, for BP-A1 and BPA2, respectively. *N^1^*-acetyl-*N^5^*, *N^10^-di-p*-coumaroylspermidine (*m*/*z* 478), *N^1^*, *N^5^*-*di*-*p*-coumaroyl-*N^10^*-caffeoylspermidine (*m*/*z* 598), *N^1^*, *N^5^*-*di*-*p*-coumaroyl-*N^10^*-caffeoylspermidine (*m*/*z* 598), tetracoumaroyl spermine (*m*/*z* 785) and its isomers (*m*/*z* 785) were detected at different concentrations in all three bee pollen samples at very high levels. In bee bread samples, tetracoumaroyl spermine and its isomers were found in lower concentrations compared to bee pollen. However, bee bread samples exhibited a rich profile especially in respect to *N^1^*, *N^5^*, *N^10^*-*tri*-caffeoylspermidine (*m*/*z* 630), *N^1^*, *N^5^*, *N^10^*-*tri*-*p*-coumaroylspermidine (*m*/*z* 582) and *N^1^*, *N^5^*, *N^10^*-*tri*-*p*-coumaroylspermidine (isomer) (*m*/*z* 582), which can be related to the botanical origin of the samples. In the study conducted by Urcan et al. [35], it was found that bee bread with origin in pollens from plant families such as Asteraceae and Fabaceae have a predominance of phenylamides.

The effect of in vitro gastrointestinal digestion on the stability of phenolic compounds and phenylamides is given in Table 3 and Table 4. Accordingly, after the oral phase, the bioactive compounds in the BP-A1, BP-A2 and BP-A3 samples decreased by 54%, 92% and 7%, respectively. Among the bee bread samples, the highest decrease was in BB-A3 (25%), followed by BB-A2 (17%) and BB-A1 (4%). Despite these decreases were statistically significant (*p* < 0.05), they were not very high for BP-A3, BB-A1 and BB-A2. This may be due to contact with the enzyme or short digestion time [38]. These reductions were comparable to the phenolic compound results reported by Quan et al. [23] and Lucas-Gonzalez et al. [38] for different food matrices.

Compared with the oral phase, the phenolic content in all samples was significantly (*p* < 0.05) reduced at the end of the gastric phase. While the content of isorhamnetin-*O*-pentosyl-hexoside (*m*/*z* 609) in the raw BP-A1 and BP-A2 samples was 0.22 and 0.07 mg/g, respectively, its concentration in both samples decreased to a minimum of 0.05 mg/g after the gastric phase. A significant decreasing trend was also observed in the BP-A3 phenolic and phenylamide compounds when comparing to the other two bee pollen samples. For the bee bread samples, a similar decrease was observed on the phenolic content at the end of the gastric phase, presenting bee bread, except for BB-A3 sample, a richer phenolic content when comparing to the bee pollen samples. As mentioned by some authors earlier [20,26], the decrease in the concentration of the phenolic compounds and phenylamides during bee pollen and bee bread digestion could be explained by the interaction with other food ingredients, causing changes in their molecular weights, solubility and chemical structures. Moreover, hydrolysis of the released compounds, mainly as a result of acidic pH (pH 2–3) and enzymatic activity, may have another important effect on this decrease [22].

In the intestinal phase, Table 3 and Table 4, there was a slight increase in the phenolic content of BP-A1 and BP-A3, while no change was observed in the BP-A2 content. On the other hand, there was a significant decrease in the phenolic content of all bee bread samples, especially in BB-A1 and BB-A2. Partial decreases and increases in the concentration of phenolic compounds after the intestinal phase could be attributed to the instability of these compounds under alkaline conditions and their possible interactions with other food components like protein, lipid or fiber, as in the gastric phase [19,20]. Besides, phenylamides are known to be completely digested and absorbed in the intestine [39], and our findings show that the identified phenylamides in the samples were completely digested after the intestinal phase. In the calculation based on the total amount of phenolic compounds found in the samples, BP-A2 had the highest bioaccessibility score with a rate of 57%, followed by BB-A1 (35%) > BB-A2 (28%) > BP-A1 (24%) > BP-A3 (17%) > BB-A3 (11%). After in vitro gastrointestinal digestion, the available quantity and diversity of compounds varied significantly (*p* < 0.05) depending on the tested sample.

The differences observed in the bioavailability values are due to the diversified chemical structures found in both phenolic and phenylamide compounds, which can range from simple to highly polymerized molecules. Different conditions (enzymatic activity and/or pH changes) of gastrointestinal digestion cause various changes in the phenol structure such as hydroxylation, methylation, isoprenylation, dimerization and glycosylation, and consequently affect the stability and bioaccessibility of these compounds [19,38].

### 3.4. Antioxidant Capacity

Antioxidant capacities of undigested and digested bee pollen and bee bread samples were measured using DPPH free radicals scavenging activity and reducing power assays. Among raw bee pollen samples, BP-A3 showed both the highest free radical scavenging (EC_50_: 0.14 ± 0.0 mg/mL) and reducing power activity (5.0 ± 0.1 mg GAE/g), much better than BP-A1 and BP-A2 (Figure 5). In raw bee bread samples, the behavior did not change much between them, with BB-A2 exhibiting the highest free radical scavenging activity with a value of 0.23 ± 0.0 mg/mL and reducing power activity with 5.6 ± 0.3 mg GAE/g, which is in accordance with previous studies that reported that bee pollen and bee bread may have different antioxidant activity [4,40].

At the end of in vitro gastrointestinal digestion, a decrease of 35–85% in free radicals scavenging capacity and of 47–76% in reducing power occurred for bee pollen samples, while in bee bread samples, there was a decrease of 69–74% in free radicals scavenging capacity and of 33–50% in reducing power activity (Figure 5). In addition, all bee pollen samples showed a steadily decreasing trend in terms of DPPH scavenging activity after each digestion phase. The same situation was observed in reducing power activity, except in the intestinal phase for BP-A3. As expected, the BP-A3, which has the highest antioxidant activity in the undigested samples, also showed the highest activity at the end of digestion, for both tests. These differences in bee pollen samples may be related to the high content released from samples and slowing down in the intestinal phase or the degradation of the released compounds under conditions in the intestinal phase [19,21].

All bee bread samples showed a higher antioxidant capacity at the end of the intestinal phase compared to the gastric phase. Several other studies, with different food matrices, report the same behavior during digestion, emphasizing that antioxidant activity was higher in the intestinal phase compared to the gastric phase [13,22]. A comparison between bee pollen and bee bread revealed that, for both antioxidant assays, bee bread showed higher antioxidant activity at the end of digestion than bee pollen.

The correlation between bioactive compounds and antioxidant capacity in bee pollen and bee bread was evaluated (Appendix A). Accordingly, the DPPH value showed significant and very strong positive correlation with both TPC (r = 0.799; *p* < 0.01) and TFC (r = 0.784; *p* < 0.01). The reducing power value was positively correlated strongly with TPC (r = 0.743; *p* < 0.01), while it was moderately correlated with TFC (r = 0.562; *p* < 0.01). The moderate correlation of reducing power with TFC compared to TPC could be attributed to the effect of digestion on the structure of flavonoids, and this is clearly seen in Figure 2. Besides, these results are supported by the findings given above, where samples with high TPC and TFC generally exhibited higher antioxidant activity. Correlation analysis indicates that the antioxidant compounds in bee pollen and bee bread are not only potent radical scavengers but also good reducing agents.

## 4. Conclusions

Bee pollen and bee bread were analyzed for botanical origin, bioactive compounds, and antioxidant capacity using an in vitro simulated gastrointestinal digestion model. The palynological analysis revealed that Plantaginaceae, Asteraceae, Fabaceae and Fagaceae were the dominant plant families. The in vitro digestion results showed that the bioactive compounds in bee pollen and bee bread generally tended to decrease throughout digestion with some exceptions. Besides, the decrease or increase in phenolic compounds in both bee products affected their antioxidant activities. Especially, the gastric phase was found to be an important factor on the bioavailability of bioactive compounds. Moreover, bee bread was more digestible than bee pollen. Considering all these findings, both bee products have an important potential for the food industry. Besides the use in the food industry, bee pollen and bee bread can find a number of applications in different fields, thanks to their phenolic compound content.

## Figures and Tables

**Figure 1 antioxidants-10-00651-f001:**
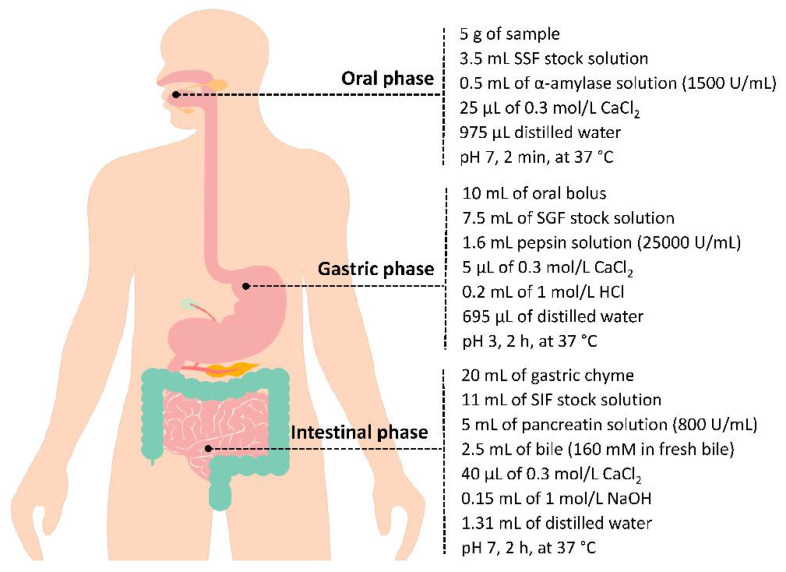
Overview of a simulated in vitro digestion method. Enzyme activities are in units per mL of the final digestion mixture at each corresponding digestion phase.

**Figure 2 antioxidants-10-00651-f002:**
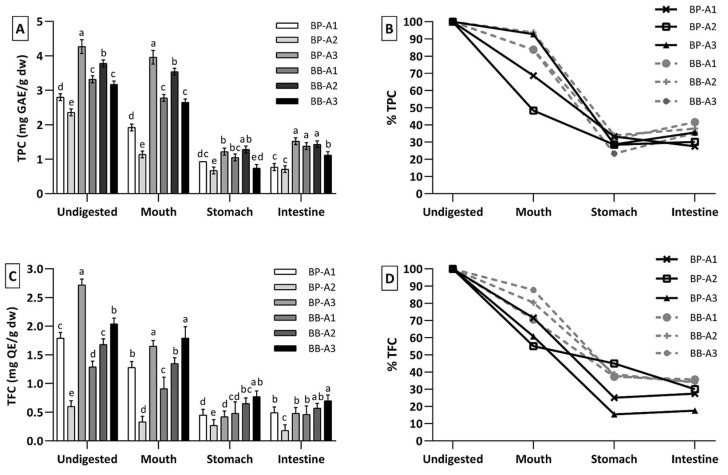
(**A**) The total phenolic content (TPC), (**B**) percentage change in TPC in each digestion phase, (**C**) the total flavonoid content (TFC) and (**D**) percentage change in TFC after each digestion phase. Within each group (undigested, mouth, stomach and intestine), different letters (a–e) mean significantly different (*p* < 0.05).

**Figure 3 antioxidants-10-00651-f003:**
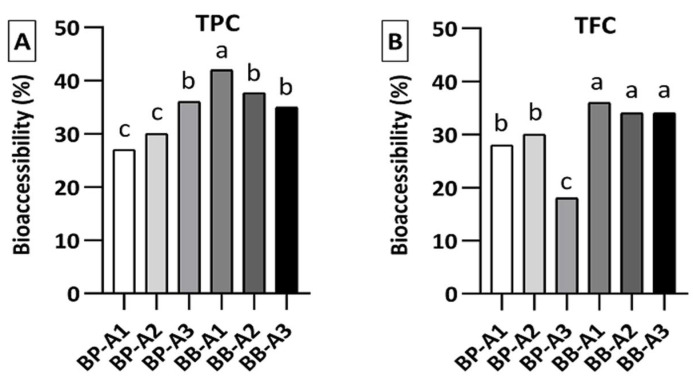
Bioaccessibility degree for (**A**) total phenolic content (TPC) and (**B**) the total flavonoid content (TFC). Within each graph (TPC or TFC), different letters (a–c) mean significantly different (*p* < 0.05).

**Figure 4 antioxidants-10-00651-f004:**
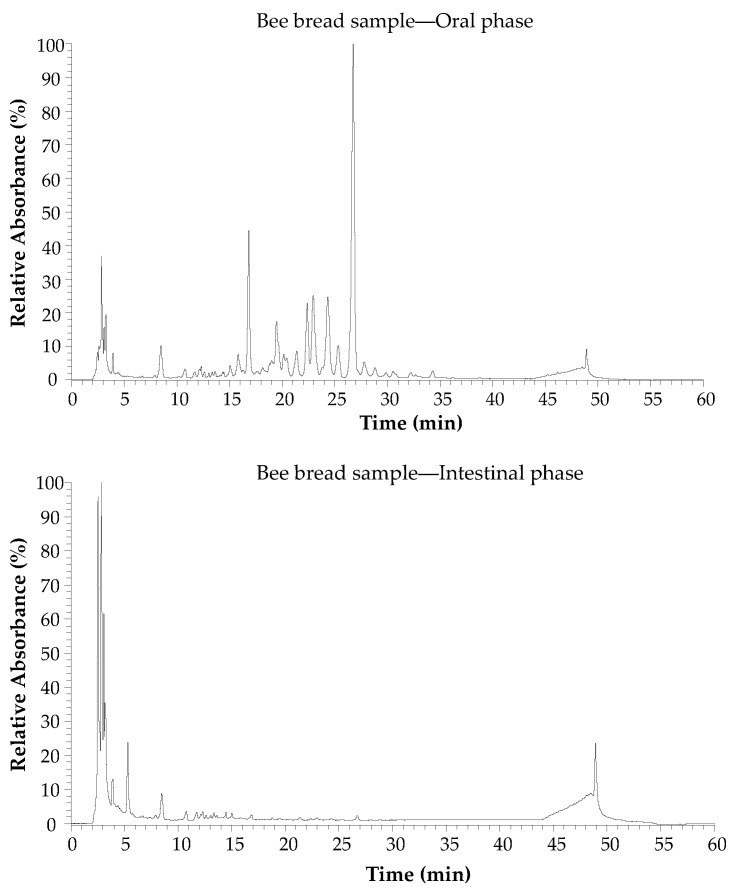
Chromatographic profile of bee bread sample obtained at 280 nm by LC/DAD/ESI-MS^n^.

**Figure 5 antioxidants-10-00651-f005:**
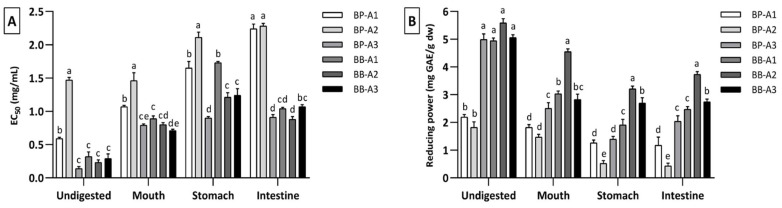
Antioxidant capacities of bee pollen and bee bread samples: (**A**) DPPH free radical scavenging and (**B**) reducing power activity of the undigested and digested samples after each in vitro digestion phase. Within each group (undigested, mouth, stomach and intestine), different letters (a–e) mean significantly different (*p* < 0.05).

**Table 1 antioxidants-10-00651-t001:** Relative frequency (%) of pollen types in bee pollen and bee bread samples.

		Relative Frequency (%) of Pollen Types					
Family	Pollen Types	BP-A1 ^a^	BP-A2 ^b^	BP-A3	BB-A1 ^a^	BB-A2 ^b^	BB-A3
**Amaranthaceae**	*Chenopodium* sp.	—	1.60	—	—	—	1.36
**Apiaceae**	*Eryngium*	1.02	—	—	—	—	0.14
	*Thapsia vilosa*	0.11	6.75	—	—	—	—
**Asteraceae**	*Centaurea* sp.	6.21	1.60	—	1.90	—	—
	*Crepis capillaris*	11.17	59.84	—	0.76	0.80	1.64
**Boraginaceae**	*Echium* sp.	8.80	0.57	0.72	4.19	1.50	—
	*Pentaglotis sempervirens*	—	—	—	1.02	—	—
**Brassicaceae**	*Raphanus raphanistrum*	—	—	3.79	0.38	0.10	1.91
**Campanulaceae**	*Jasione montana*	—	—	—	0.13	3.40	21.96
**Crassulaceae**	*Sedum* sp.	—	—	0.21	0.51	3.30	0.82
**Ericaceae**	*Erica* sp.	—	—	9.03	—	—	—
**Fabaceae**	*Lotus* sp.	0.11	—	—	1.14	—	—
	*Cytisus striatus*	—	0.11	47.69	6.22	6.50	19.65
	*Trifolium* sp.	5.98	0.23	1.74	12.18	6.50	—
**Fagaceae**	*Castanea sativa*	—	—	—	25.38	47.50	25.51
	*Quercus* sp.	—	0.34	—	—	—	6.96
**Lamiaceae**	*Lavandula* sp.	—	—	—	0.38	1.10	3.27
**Myrtaceae**	*Eucalyptus* sp.	—	—	24.10	—	—	—
**Papaveraceae**	—	0.34	0.23	3.49	0.25	2.20	0.55
**Plantaginaceae**	*Plantago* sp.	47.18	20.14	0.10	2.79	1.20	1.91
**Poaceae**	*Zea mays*	1.81	—	—	—	—	—
**Resedaceae**	*Sesamoides* sp. *or Reseda* sp.	—	—	1.85	0.13	0.20	—
	*Rhamnus alaternus*	—	—	0.10	1.02	—	—
**Rosaceae**	—	—	—	1.33	—	—	0.68
	*Rubus* sp.	0.56	—	2.67	37.31	22.30	10.23
**Salicaceae**	*Salix* sp.	—	—	1.33	0.51	0.80	—
**Solanaceae**	—	—	2.06	—	—	—	—
**Classification**		Multifloral					

^a, b^ The same letters represent samples collected in the same apiary. Dominant pollen (>45%); accessory pollen (15–45%) and isolated pollen (3–15%).

**Table 2 antioxidants-10-00651-t002:** Phenolic and phenylamide profile of raw bee pollen and bee bread samples.

t_R_ (min)	λ_max_ (nm)	[M-H]^-^ *m*/*z*	MS^n^ (% Base Peak)	Proposed Compound	mg/g Raw Sample					
					BP-A1	BP-A2	BP-A3	BB-A1	BB-A2	BB-A3
**6.58**	255, 349	771	MS^2^: 609 (100); MS^3^: 301 (100)	Quercetin-*O*-hexosyl-*O*-rutinoside ^a,c^	0.08 ± 0.0	ND	ND	ND	ND	ND
**7.48**	257, 353	625	MS^2^: 301 (100), 300 (99), 445 (85), 271 (18)	Quercetin-diglucoside ^a,e,h,^	ND	ND	ND	ND	ND	ND
**8.43**	272, 326sh, 353sh	639	MS^2^:271 (10), 300 (34), 315 (91), 459 (100), 477 (11), 624 (20)	Methyl herbacetin-*O*-dihexoside ^a,c,d^	0.04 ± 0.0	ND	0.31 ± 0.0	0.43 ± 0.0	0.33 ± 0.0	0.14 ± 0.0
**9.94**	265, 348	609	MS^2^: 285 (100), 429 (49)	Kaempferol-*O*-dihexoside ^a,d^	0.08 ± 0.0	ND	ND	ND	ND	ND
**10.69**	272, 326sh, 353sh	623	MS^2^: 299 (61), 300 (38), 314 (100), 315 (69), 459 (86), 477 (19)	Methyl herbacetin-*O*-rutinoside ^a,c^	ND	ND	ND	0.09 ± 0.0	0.08 ± 0.0	0.02 ± 0.0
**11.69**	255, 353	609	MS^2^: 315 (100), 314 (47), 459 (51), 300 (20)	Isorhamnetin-*O*-pentosyl-hexoside ^a,e^	0.22 ± 0.0	0.07 ± 0.0	ND	0.06 ± 0.0	0.05 ± 0.0	0.06 ± 0.0
**12.09**	265, 348	593	MS^2^: 284 (100), 285 (73), 429 (93)	Kaempferol-3-*O*-rutinoside ^a,b,f^	0.02 ± 0.0	0.00 ± 0.0	0.03 ± 0.0	0.01 ± 0.0	0.02 ± 0.0	0.01 ± 0.0
**12.62**	256, 354	463	MS^2^: 301 (100)	Quercetin-3-*O*-glucoside ^a,b,f^	0.01 ± 0.0	ND	ND	ND	ND	0.02 ± 0.0
**13.31**	256, 353	549	MS^2^: 505 (100); MS^3^: 301 (100), 300 (28), 463 (26)	Quercetin-*O*-malonyl hexoside ^a,j^	ND	ND	ND	ND	0.01 ± 0.0	0.07 ± 0.0
**13.61**	270	477	MS^2^: 315 (100), 462 (42), 300 (14); MS^3^: 300 (100)	Methyl herbacetin-3-*O*-hexoside ^a,c^	ND	ND	ND	0.04 ± 0.0	ND	ND
**14.06**	265, 347	447	MS^2^: 285 (100), 284 (80)	Kaempferol-*O*-hexoside ^a,k^	ND	ND	ND	ND	0.03 ± 0.0	0.16 ± 0.0
**14.18**	254, 347	447	MS^2^: 301 (100)	Quercetin-3-*O*-rhamnoside^a,b,e^	ND	ND	0.81 ± 0.0	ND	ND	ND
**14.31**	254, 355	477	MS^2^: 314 (100), 315 (45)	Isorhamnetin-*O*-hexoside ^a,k^	ND	ND	ND	ND	0.01 ± 0.0	0.01 ± 0.0
**14.44**	255, 354	563	MS^2^: 519 (100); MS^3^: 315 (100)	Isorhamnetin-3-*O*-malonyl glucoside ^a,h^	ND	ND	0.06 ± 0.0	ND	ND	ND
**14.73**	277, 311	301	MS^2^: 283 (100), 286 (40)	Hesperetin ^a,b^	ND	ND	0.05 ± 0.0	ND	ND	ND
**14.99**	265, 345	533	MS^2^: 489 (100); MS^3^: 285 (100)	Kaempferol-*O*-malonyl hexoside ^a,h^	ND	ND	ND	ND	0.07 ± 0.0	0.09 ± 0.0
**15.39**	299, 308	436	MS^2^: 316 (100)	Di-*p*-coumaroylspermidine ^a,i^	0.72 ± 0.0	0.20 ± 0.0	ND	ND	ND	ND
**15.76**	295, 315	630	MS^2^: 468 (100), 494 (84), 358 (7); MS^3^: 332 (100)	*N^1^, N^5^, N^10^-tri*-caffeoylspermidine ^a,d,g^	ND	ND	ND	0.20 ± 0.0	0.50 ± 0.0	ND
**16.18**	264, 341	431	MS^2^: 285 (100)	Kaempferol-3-*O*-rhamnoside ^a,c^	ND	ND	0.04 ± 0.0	ND	ND	ND
**16.64**	255, 354	461	MS^2^: 314 (100), 315 (77), 299 (39)	Isorhamnetin-*O*-deoxyhexoside ^a,c^	ND	ND	0.31 ± 0.0	ND	ND	ND
**16.75**	296, 319	630	MS^2^: 468 (100), 494 (86), 358 (7); MS^3^: 332 (100)	*N^1^, N^5^, N^10^-tri*-caffeoylspermidine ^a,d,g^	0.43 ± 0.0	0.08 ± 0.0	ND	1.53 ± 0.0	4.15 ± 0.0	0.51 ± 0.0
**17.57**	293, 314	644	MS^2^: 358 (11), 482 (11), 508 (100); MS^3^: 332 (27), 358 (100), 372 (49)	*N^1^*-feruloyl-*N^5^, N^10^*-dicaffeoylspermidine ^a,d,e^	ND	ND	ND	ND	0.04 ± 0.0	ND
**18.09**	295, 311	614	MS^2^: 494 (25), 478 (100), 452 (69), 358 (20)	*N^1^-p*-coumaroyl-*N^5^, N^10^*-dicaffeoylspermidine ^a,e^	0.21 ± 0.0	ND	0.38 ± 0.0	0.16 ± 0.0	0.13 ± 0.0	0.07 ± 0.0
**18.18**	299, 308	478	MS^2^: 358 (100), 332 (12), 145 (5)	*N^1^*-acetyl-*N^5^, N^10^-di-p*-coumaroylspermidine ^a,f^	2.83 ± 0.1	0.69 ± 0.0	ND	ND	ND	ND
**18.74**	295, 311	614	MS^2^: 478 (100), 468 (20), 452 (68), 342(5)	*N^1^*-*p*-coumaroyl-*N^5^, N^10^*-dicaffeoylspermidine (isomer) ^a,e^	ND	ND	ND	0.06 ± 0.0	0.20 ± 0.0	ND
**19.34**	295, 311	614	MS^2^: 494 (24), 478 (100), 452 (76), 358 (22)	*N^1^-p*-coumaroyl-*N^5^, N^10^*-dicaffeoylspermidine (isomer) ^a,e^	ND	ND	1.36 ± 0.0	0.76 ± 0.0	1.98 ± 0.0	0.89 ± 0.0
**20.02**	295, 318	644	MS^2^: 358 (8), 482 (75), 508 (100); MS^3^: 332 (27), 358 (100), 372 (49)	*N^1^*-feruloyl-*N^5^, N^10^*-dicaffeoylspermidine (isomer) ^a,d,g^	ND	ND	ND	ND	0.18 ± 0.0	ND
**20.27**	295, 310	598	MS^2^: 478 (46), 462 (100), 452 (46), 342 (14)	*N^1^, N^5^-di-p*-coumaroyl-*N^10^*-caffeoylspermidine ^a,e^	ND	ND	ND	0.22 ± 0.0	0.08 ± 0.0	ND
**21.23**	295, 310	582	MS^2^: 462 (100), 436 (9), 342 (7)	*N^1^, N^5^, N^10^-tri*-*p*-coumaroylspermidine ^a,e^	ND	0.13 ± 0.0	ND	0.69 ± 0.0	1.02 ± 0.0	0.42 ± 0.0
**21.33**	254, 268sh, 348	285	MS^2^: 285 (100)	Luteolin ^a,b^	ND	ND	1.00 ± 0.0	ND	ND	ND
**22.22**	294, 309	598	MS^2^: 462 (100), 478 (39), 452 (34), 342 (14)	*N^1^, N^5^-di-p*-coumaroyl-*N^10^*-caffeoylspermidine ^a,d,g^	0.21 ± 0.0	0.48 ± 0.0	1.16 ± 0.0	1.40 ± 0.0	2.12 ± 0.0	0.85 ± 0.0
**22.73**	295, 310	582	MS^2^: 462 (100), 436 (9), 342 (7)	*N^1^, N^5^, N^10^-tri*-*p*-coumaroylspermidine ^a,e^	ND	0.16 ± 0.0	ND	1.82 ± 0.0	2.84 ± 0.0	1.68 ± 0.0
**22.77**	295, 310	598	MS^2^: 342 (13), 452 (32), 462 (100), 478 (37)	*N^1^, N^5^-di-p*-coumaroyl-*N^10^*-caffeoylspermidine (isomer) ^a,e^	ND	ND	2.61 ± 0.1	ND	ND	ND
**24.12**	295, 310	582	MS^2^: 462 (100), 436 (9), 342 (6)	*N^1^, N^5^, N^10^-tri*-*p*-coumaroylspermidine ^a,e^	0.14 ± 0.0	0.18 ± 0.0	0.61 ± 0.0	2.13 ± 0.0	3.20 ± 0.0	0.93 ± 0.0
**25.08**	295, 310	582	MS^2^: 462 (100), 436 (9), 342 (7)	*N^1^, N^5^, N^10^-tri*-*p*-coumaroylspermidine ^a,e^	ND	ND	0.20 ± 0.0	0.77 ± 0.0	1.12 ± 0.1	0.34 ± 0.0
**26.47**	295, 305	582	MS^2^: 342 (100), 436 (9), 462 (100)	*N^1^, N^5^, N^10^*-tri-*p*-coumaroylspermidine (isomer) ^a,e^	0.55 ± 0.0	0.29 ± 0.0	6.28 ± 0.0	8.47 ± 0.0	15.23 ± 0.1	5.33 ± 0.0
**26.92**	270	785	MS^2^: 665 (100), 545 (14), 639 (13); MS^3^: 545 (100)	Tetracoumaroyl spermine ^a,l^	ND	3.34 ± 0.0	ND	ND	ND	ND
**27.52**	295, 308	612	MS^2^: 492 (100); MS^3^: 372 (100), 449 (24)	Feruloyl dicoumaroyl spermidine ^a,l^	ND	ND	0.93 ± 0.0	0.32 ± 0.0	0.70 ± 0.0	ND
**27.67**	271	315	MS^2^: 300 (100); MS^3^: 272 (100), 255 (54), 165 (26)	Herbacetin-methyl-ether ^a^	ND	ND	ND	ND	ND	0.11 ± 0.0
**28.55**	280, 307sh	785	MS^2^: 665 (100), 545 (13), 639 (13); MS^3^: 545 (100)	Tetracoumaroyl spermine (isomer) ^a,l^	0.90 ± 0.0	8.25 ± 0.1	0.76 ± 0.0	0.42 ± 0.0	0.31 ± 0.0	ND
**28.77**	266, 365	285	MS^2^: 285 (100)	Kaempferol ^a,b^	ND	ND	ND	ND	ND	0.28 ± 0.0
**29.28**	277, 310sh	785	MS^2^: 665 (100), 545 (13), 639 (13); MS^3^: 545 (100)	Tetracoumaroyl spermine (isomer) ^a,l^	ND	0.83 ± 0.1	ND	ND	ND	ND
**29.57**	295, 318	672	-	Polyamide derivative ^a,e^	ND	ND	1.70 ± 0.0	ND	ND	ND
**30.28**	289, 306sh	785	MS^2^: 665 (100), 545 (13), 639 (13); MS^3^: 545 (100)	Tetracoumaroyl spermine (isomer) ^a,l^	0.74 ± 0.0	9.11 ± 0.1	ND	0.50 ± 0.0	ND	ND
**31.92**	293, 310	785	MS^2^: 665 (100), 545 (13), 639 (13); MS^3^: 545 (100)	Tetracoumaroyl spermine (isomer) ^a,l^	0.50 ± 0.0	6.51 ± 0.1	ND	0.47 ± 0.0	ND	0.36 ± 0.0
**33.97**	299, 310	785	MS^2^: 665 (100), 545 (13), 639 (13); MS^3^: 545 (100)	Tetracoumaroyl spermine (isomer) ^a,l^	0.70 ± 0.0	8.49 ± 0.1	ND	0.86 ± 0.0	0.36 ± 0.1	ND
**Total amount of phenolic compounds**					0.5	0.1	2.6	0.6	0.6	1.0
**Total amount of phenylamides**					7.9	38.7	15.9	20.8	34.2	11.4

Confirmed with: ^a^ MS^n^ fragmentation; ^b^ Standard; References: ^c^ [10]; ^d^ [6]; ^e^ [9]; ^f^ [27]; ^g^ [28]; ^h^ [29]; ^i^ [30]; ^j^ [31]; ^k^ [32]; ^l^ [33]. Values expressed as mg of each compound/g sample. ND = not detected.

**Table 3 antioxidants-10-00651-t003:** Phenolic and phenylamide compounds of the bee pollen samples obtained after each in vitro digestion phase.

Compound	Mouth			Stomach			Intestine		
	BP-A1	BP-A2	BP-A3	BP-A1	BP-A2	BP-A3	BP-A1	BP-A2	BP-A3
**Quercetin-*O*-hexosyl-*O*-rutinoside**	0.03 ± 0.0	ND	ND	ND	ND	ND	ND	ND	ND
Quercetin-diglucoside	ND	ND	0.03 ± 0.0	ND	ND	0.04 ± 0.0	ND	ND	0.05 ± 0.0
Methyl herbacetin-*O*-dihexoside	0.03 ± 0.0	ND	0.32 ± 0.0	ND	ND	0.17 ± 0.0	ND	ND	0.22 ± 0.0
Kaempferol-*O*-dihexoside	0.05 ± 0.0	ND	ND	ND	ND	ND	ND	ND	ND
Isorhamnetin-*O*-pentosyl-hexoside	0.12 ± 0.0	0.06 ± 0.0	ND	0.05 ± 0.0	0.05 ± 0.0	ND	0.09 ± 0.0	0.04 ± 0.0	ND
Kaempferol-3-*O*-rutinoside	0.01 ± 0.0	0.00 ± 0.0	0.03 ± 0.0	ND	ND	0.01 ± 0.0	0.01 ± 0.0	ND	0.01 ± 0.0
Quercetin-3-*O*-glucoside	0.01 ± 0.0	ND	ND	ND	ND	ND	0.01 ± 0.0	ND	ND
Quercetin-3-*O*-rhamnoside	ND	ND	0.77 ± 0.0	ND	ND	0.15 ± 0.0	ND	ND	0.14 ± 0.0
Isorhamnetin-3-*O*-malonyl glucoside	ND	ND	0.03 ± 0.0	ND	ND	0.01 ± 0.0	ND	ND	ND
Hesperetin	ND	ND	0.05 ± 0.0	ND	ND	ND	ND	ND	ND
Di-*p*-coumaroylspermidine	0.77 ± 0.0	0.03 ± 0.0	ND	ND	ND	ND	ND	ND	ND
Kaempferol-3-*O*-rhamnoside	ND	ND	0.03 ± 0.0	ND	ND	ND	ND	ND	ND
Isorhamnetin-*O*-deoxyhexoside	ND	ND	0.31 ± 0.0	ND	ND	0.02 ± 0.0	ND	ND	0.02 ± 0.0
***N^1^, N^5^, N^10^-tri*-caffeoylspermidine**	0.26 ± 0.0	0.04 ± 0.0	ND	ND	ND	ND	ND	ND	ND
***N^1^*-acetyl-*N^5^*, *N^10^-di-p-*coumaroylspermidine**	2.01 ± 0.1	0.35 ± 0.0	ND	ND	ND	ND	ND	ND	ND
***N^1^-p*-coumaroyl-*N^5^*, *N^10^*-dicaffeoylspermidine**	0.14 ± 0.0	ND	0.39 ± 0.0	ND	ND	ND	ND	ND	ND
*N^1^-p*-coumaroyl-*N^5^*, *N^10^*-dicaffeoylspermidine (isomer)	ND	ND	1.16 ± 0.0	ND	ND	ND	ND	ND	ND
***N^1^*, *N^5^-di-p*-coumaroyl-*N^10^*-caffeoylspermidine**	0.10 ± 0.0	0.22 ± 0.0	1.17 ± 0.0	ND	ND	ND	ND	ND	ND
*N^1^*, *N^5^-di-p*-coumaroyl-*N^10^*-caffeoylspermidine (isomer)	ND	ND	1.97 ± 0.1	ND	ND	ND	ND	ND	ND
Luteolin	ND	ND	0.99 ± 0.0	ND	ND	ND	ND	ND	ND
***N^1^*, *N^5^*, *N^10^-tri-p*-coumaroylspermidine**	ND	0.11 ± 0.0	0.56 ± 0.0	ND	ND	ND	ND	ND	ND
***N^1^*, *N^5^*, *N^10^-tri-p*-coumaroylspermidine (isomer)**	0.43 ± 0.0	0.22 ± 0.0	6.31 ± 0.1	ND	ND	ND	ND	ND	ND
Tetracoumaroyl spermine	ND	ND	ND	ND	0.18 ± 0.0	ND	ND	ND	ND
Feruloyl dicoumaroyl spermidine	ND	ND	0.75 ± 0.0	ND	ND	ND	ND	ND	ND
Tetracoumaroyl spermine (isomer)	ND	0.95 ± 0.0	ND	ND	ND	ND	ND	ND	ND
Diferuloyl coumarouyl spermidine	0.75 ± 0.0	ND	ND	ND	ND	ND	ND	ND	ND
Polyamide derivative	ND	ND	3.12 ± 0.2	ND	ND	ND	ND	ND	ND
Tetracoumaroyl spermine (isomer)	ND	ND	ND	ND	0.36 ± 0.0	ND	ND	ND	ND
Tetracoumaroyl spermine (isomer)	ND	ND	ND	ND	0.03 ± 0.0	ND	ND	ND	ND
Tetracoumaroyl spermine (isomer)	ND	0.99 ± 0.0	ND	ND	0.36 ± 0.0	ND	ND	ND	ND
Tetracoumaroyl spermine (isomer)	ND	ND	ND	ND	0.23 ± 0.0	ND	ND	ND	ND
**Total amount of phenolic compounds**	0.2	0.1	2.6	0.1	0.0	0.4	0.1	0.0	0.4
**Total amount of phenylamides**	3.7	2.9	14.7	0.0	1.2	0.0	0.0	0.0	0.0

Values expressed as milligram of each compound/g sample. ND = not detected.

**Table 4 antioxidants-10-00651-t004:** Phenolic and phenylamide compounds of the bee bread samples obtained after each in vitro digestion phase.

Compound	Mouth			Stomach			Intestine		
	BB-A1	BB-A2	BB-A3	BB-A1	BB-A2	BB-A3	BB-A1	BB-A2	BB-A3
Methyl herbacetin-*O*-dihexoside	0.41 ± 0.0	0.33 ± 0.0	0.07 ± 0.0	0.22 ± 0.0	0.16 ± 0.0	0.06 ± 0.0	0.16 ± 0.0	0.12 ± 0.0	0.04 ± 0.0
Methyl herbacetin-*O*-rutinoside	0.08 ± 0.0	0.08 ± 0.0	0.01 ± 0.0	0.05 ± 0.0	0.05 ± 0.0	ND	0.04 ± 0.0	0.04 ± 0.0	ND
Isorhamnetin-*O*-pentosyl hexoside	0.05 ± 0.0	0.06 ± 0.0	0.04 ± 0.0	0.02 ± 0.0	ND	0.03 ± 0.0	0.02 ± 0.0	ND	0.04 ± 0.0
Kaempferol-3-*O*-rutinoside	0.01 ± 0.0	0.01 ± 0.0	0.01 ± 0.0	0.00 ± 0.0	0.00 ± 0.0	0.01 ± 0.0	ND	0.00 ± 0.0	0.00 ± 0.0
Quercetin-3-*O*-glucoside	ND	ND	0.01 ± 0.0	ND	ND	0.00 ± 0.0	ND	ND	ND
Quercetin-*O*-malonyl hexoside	ND	0.02 ± 0.0	0.02 ± 0.0	ND	ND	ND	ND	ND	ND
Methyl herbacetin-3-*O*-hexoside	0.02 ± 0.0	ND	ND	0.01 ± 0.0	ND	ND	ND	ND	ND
Kaempferol-*O*-hexoside	ND	0.00 ± 0.0	0.01 ± 0.0	ND	ND	ND	ND	ND	ND
Isorhamnetin-*O*-hexoside	ND	0.01 ± 0.0	ND	ND	ND	ND	ND	ND	ND
**Kaempferol-*O*-malonyl hexoside**	ND	0.03 ± 0.0	0.03 ± 0.0	ND	0.01 ± 0.0	0.03 ± 0.0	ND	0.01 ± 0.0	0.02 ± 0.0
***N^1^, N^5^, N^10^-tri*-caffeoylspermidine**	1.84 ± 0.1	4.14 ± 0.2	0.40 ± 0.0	0.07 ± 0.0	0.09 ± 0.0	ND	ND	ND	ND
***N^1^*-feruloyl-*N^5^*, *N^10^*-dicaffeoylspermidine**	ND	0.06 ± 0.0	ND	ND	ND	ND	ND	ND	ND
***N^1^*-feruloyl-*N^5^*, *N^10^*-dicaffeoylspermidine (isomer)**	ND	0.17 ± 0.0	ND	ND	ND	ND	ND	ND	ND
***N^1^-p*-coumaroyl-*N^5^, N^10^*-dicaffeoylspermidine**	0.21 ± 0.0	0.13 ± 0.0	0.06 ± 0.0	ND	ND	ND	ND	ND	ND
***N^1^-p*-coumaroyl-*N^5^, N^10^*-dicaffeoylspermidine (isomer)**	0.50 ± 0.0	1.48 ± 0.0	0.69 ± 0.0	ND	0.03 ± 0.0	ND	ND	ND	ND
***N^1^*, *N^5^*-*di-p*-coumaroyl-*N^10^*-caffeoylspermidine**	0.30 ± 0.0	0.07 ± 0.0	0.66 ± 0.0	ND	ND	ND	ND	ND	ND
***N^1^*, *N^5^*-*di-p*-coumaroyl-*N^10^*-caffeoylspermidine (isomer)**	1.23 ± 0.0	1.79 ± 0.1	ND	0.07 ± 0.0	ND	ND	ND	ND	ND
***N^1^, N^5^, N^10^-tri-p-*coumaroylspermidine**	12.25 ± 0.1	19.78 ± 0.2	7.17 ± 0.1	0.42 ± 0.0	0.18 ± 0.0	ND	ND	ND	ND
Methyl herbacetin	ND	ND	0.09 ± 0.0	ND	ND	ND	ND	ND	ND
Feruloyl dicoumaroyl spermidine	0.54 ± 0.1	0.50 ± 0.1	ND	ND	ND	ND	ND	ND	ND
Tetracoumaroyl spermine	0.59 ± 0.0	0.23 ± 0.0	ND	ND	ND	ND	ND	ND	ND
Tetracoumaroyl spermine (isomer)	0.69 ± 0.0	ND	ND	ND	ND	ND	ND	ND	ND
Tetracoumaroyl spermine (isomer)	0.68 ± 0.1	ND	ND	ND	ND	ND	ND	ND	ND
Tetracoumaroyl spermine (isomer)	1.14 ± 0.1	ND	ND	ND	ND	ND	ND	ND	ND
**Total amount of phenolic compounds**	0.6	0.6	0.3	0.3	0.2	0.1	0.2	0.2	0.1
**Total amount of phenylamides**	19.9	28.4	8.9	0.5	0.3	0.0	0.0	0.0	0.0

Values expressed as milligram of each compound/g sample. ND = not detected.

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
