# Peer review of "Assessment of Bioactive Compounds under Simulated Gastrointestinal Digestion of Bee Pollen and Bee Bread: Bioaccessibility and Antioxidant Activity"

_antioxidants, 2021, doi:10.3390/antiox10050651_

Round 1

Reviewer 1 Report

The study is very well designed, the manuscript is adequate for a possible publication in the Antioxidants journal. The high amount of data, clearly presented, methods thoroughly described make this paper a possible highly cited paper in the field. My recommendation is to Accept

Author Response

Thanks to the positive comments made by the reviewer.

Reviewer 2 Report

Dear Authors, you should address the minor comments highlighted across the text, Tables and Figures.

Author Response

The manuscript is now revised taking in considerations the proposals of the reviewers. Additional, and to avoid similarities, we rephrased section 2.6 as suggested by the editor. In attachment you can find the specific actions regarding each reviewer's comments.We would like to thank the reviewers for their valuable comments, suggestions, and contributions to this study.

Reviewer 3 Report

The paper title Assessment of bioactive compounds under simulated gastrointestinal digestion of bee pollen and bee bread: bioaccessibility and antioxidant activity made by Volkan Aylanc, Andreia Tomás, Paulo Russo-Almeida, Soraia I. Falcão, Miguel Vilas-Boas, is a very good work, very well presented and with important conclusions.

Only some minor revision before acceptance

 In section 2.4. Phenolic compounds extraction, more detail s need, for example:

“ …2 g of each sample was mixed with EtOH/H2O (80:20, v/v),..” the volume of EtOH/H2O solution used to diluted the pollen must be indicated.

“…evaporated at 40 °C.” in a rotavapor? If Yes indicate the equipment and their references

 “Lyophilized” – indicate equipment and their reference

In section 2.5.2. Total flavonoid content

“After 30 min at room temperature, the absorbance was read at 415 nm” miss “nm using 126 a spectrophotometer (Analytikijena 200–2004 spectrophotometer, Analytik Jena, Ger-127 many)” if it was another equipment please specify

Table 1 . Heterofloral must be replaced by multifloral

Supplementary material

The unit in the table are not in the same format of the manuscript. Please revise all.

Table S2 in column  “Significance” please use “Significance level”

Author Response

(The authors gave the same response as above.)
